# Age and Density of Mated Females Affect Dispersal Strategies in Spider Mite *Tetranychus ludeni* Zacher

**DOI:** 10.3390/insects15060387

**Published:** 2024-05-27

**Authors:** Peng Zhou, Xiong Z. He, Chen Chen, Qiao Wang

**Affiliations:** 1Anhui Key Laboratory of Biodiversity Research and Ecological Protection in Southwest Anhui Province, School of Life Sciences, Anqing Normal University, Anqing 246133, China; p.zhou@aqnu.edu.cn (P.Z.); chenchen230628@163.com (C.C.); 2School of Agriculture and Environment, Massey University, Private Bag 11222, Palmerston North 4414, New Zealand; x.z.he@massey.ac.nz

**Keywords:** age-specific reproduction, dispersal distance, dispersal probability, population density, resource competition

## Abstract

**Simple Summary:**

The European native spider mite *Tetranychus ludeni* Zacher (Acari: Tetranychidae) is an invasive species which attacks many economically important crops. Investigations into how different factors affect its dispersal probability and distance help better understand its population management and enable researchers to perform invasion evaluations. Here, we tested the effect of age and density on dispersal probability and distance. We show that older females that are capable of producing more eggs within 24 h were more likely to disperse and moved longer distances than younger ones with fewer eggs. Older females spread most of their eggs out of their natal habitats and over longer distances, which reduced competition and increased offspring fitness. Our results also indicate that females exhibited significantly increased dispersal probability and distance with an increase in population density to avoid crowding. The synchronization of dispersal and reproduction, along with the positive density-dependent dispersal strategy, may facilitate the habitat colonization and invasion speed of *T. ludeni*.

**Abstract:**

The dispersal strategies of a species can affect its invasion success. Investigations into the dispersal strategies of invasive species in relation to different factors help improve our understanding of invasion mechanisms and provide knowledge for population management and invasion evaluation. *Tetranychus ludeni* Zacher (Acari: Tetranychidae) is an invasive species which is native to Europe but is now cosmopolitan. Here, we examined the effects of age and density on dispersal in mated females. Our results show that older females that are capable of producing more eggs within 24 h were more likely to disperse and moved longer distances than younger ones with fewer eggs. Older females spread most of their eggs out of their natal habitats and over longer distances, which reduced competition and increased offspring fitness. Females exhibited significantly increased dispersal probability and distances with an increase in population density to avoid crowding. The synchronization of dispersal and reproduction, along with the positive density-dependent dispersal strategy, may facilitate the habitat colonization and invasion speed of *T. ludeni*.

## 1. Introduction

Investigations into how different factors affect dispersal probability and distance help better understand the dispersal process [1], the knowledge of which is important for population management and invasion evaluation [2,3,4,5,6]. Population density and individual age are two important factors that can influence dispersal [1,3]. However, previous studies have mainly focused on how these factors modify dispersal probability [3,7,8,9,10,11,12]. The effect of density and age on dispersal distance and subsequent reproduction is still poorly understood.

Many species exhibit age-specific dispersal due to the variance in reproductive costs or pressures with age [4]. Females at the age of high fecundity may have a higher dispersal probability and a longer dispersal distance over which to spread their eggs [13,14,15,16]. The availability of resources is the major factor affecting immature fitness, and thus, mothers are expected to disperse and select habitats with sufficient resource supply or less resource competition, thereby increasing offspring fitness [1,11,17,18,19,20,21]. The spreading of offspring over habitats of high quality may be particularly important for species whose immatures have limited dispersal ability and no parental care. Although many studies have investigated age-specific dispersal and reproduction, few examine the subsequent offspring competition intensity and fitness. Understanding age-specific dispersal strategies in relation to reproduction and offspring fitness is essential, because the reproductive output of dispersers at the age of dispersal can affect establishment success and population growth at new resident sites [22,23,24,25,26].

High population density often leads to intense resource competition, reducing individual fitness [3,27,28]. Increasing density would potentially raise dispersal probability [2,3,9,29] over longer distances [30,31]. However, individuals living in high densities may also experience benefits; for example, studies from different taxa showed that living in a group can dilute predation risk and increase foraging efficiency [3,32], and thus, individuals may tend to live in habitats with high densities. It is predicted that density-dependent dispersal in animals depends on the balance of the costs and benefits of group living, which warrants species-specific investigations.

The European native spider mite *Tetranychus ludeni* Zacher (Acari: Tetranychidae) has now invaded all continents except Antarctica [33,34]. It is a haplodiploid species whose virgin females produce haploid sons and whose mated females produce haploid sons and diploid daughters [35,36]. *T. ludeni* attacks over 300 host species, including many economically important crops such as beans, eggplant, hibiscus, apple, pumpkin, and other cucurbitaceous plants [33,37,38,39]. Spider mites have a relatively short life cycle and high fecundity. They build up populations rapidly and exhaust resources very fast, and dispersal is common, especially in overcrowded and food-depleted environments [2,30,40,41]. Previous research has demonstrated that high population density negatively affected the individual reproduction and population growth of *T. ludeni* [42,43,44]. However, whether high density would promote the dispersal probability and distance of *T. ludeni* is unknown. Adult females of spider mites usually disperse after mating [45] through ballooning and walking (or aerial and ambulatory dispersal) [9,12]. Young females disperse more frequently than older females by ballooning [9]. However, the information on age-specific ambulatory dispersal is still not clear [12]. Moreover, mated females of *T. ludeni* gradually increase their reproductive output until they are six days old [42]. So far, it is unknown whether age-specific ambulatory dispersal is synchronized with age-specific reproduction. Investigations into how age and density affect the dispersal strategies of *T. ludeni* facilitate our understanding of its invasion mechanisms and provide knowledge for population management and invasion evaluation.

In this study, we investigated the effect of age on ambulatory dispersal probability and distance and subsequent reproduction, and that of density on dispersal probability and distance, in *T. ludeni*. We hypothesized that (1) females at the age of higher fecundity would have higher dispersal probability and longer dispersal distance and produce eggs over longer distances, and (2) high density would increase dispersal probability and distance.

## 2. Materials and Methods

### 2.1. Mite Colony and Experimental Conditions

We collected *T. ludeni* adults on *Passiflora mollissima* Bailey (Malpighiales: Passifloraceae) in Palmerston North, New Zealand, and used 3- to 5-week-old common bean (*Phaseolus vulgaris* L. (Fabales: Fabaceae)) plants grown in pots to maintain the colony. We used three pots each containing 3–4 bean plants in a tray (45 cm length × 36 cm width × 1.5 cm heigh) to rear the mites. When the mite colony had built up and the bean leaves became shriveled, we introduced three new pots containing clean bean plants into the tray. We cut the mite-infested leaves, placed them on the top of the clean plants, and removed the old pots. We used leaf squares cut from the first expanded leaves of 1- to 2-week-old bean plants grown in pots for all experiments. Environmental conditions for the colony and experiments were set at 25 ± 1 °C, 40 ± 10% RH, and a 14:10 h (L/D) photoperiod. 

### 2.2. Preparation of Mated Females for Experiments

To obtain virgin males to inseminate females, we randomly selected 50 female deutonymphs from the colony and transferred them onto a clean fresh leaf square (4 cm × 4 cm) on wet cotton wool in a Petri dish (9 cm in diameter × 1 cm in height). We allowed these deutonymphs to develop into virgin adult females and lay unfertilized eggs for three days. We then removed adult females and allowed the eggs to develop into virgin male adults. We prepared a total of 46 Petri dishes for this purpose. We used 1-day-old virgin adult males to mate with females with the characteristics indicated below.

We randomly selected 50 adult females from the colony and transferred them onto a clean fresh leaf square (4 cm × 4 cm) on wet cotton wool in a Petri dish for egg laying for 24 h. We then removed the female adults and allowed eggs to develop to the quiescent deutonymphal stage. We set up approximately 230 Petri dishes for this purpose. We transferred 15~20 virgin males with the characteristics indicated in the previous paragraph to each of those Petri dishes, allowing them to stay with the newly emerged females for 5 h to ensure that all females mated at or soon after emergence (females mate immediately after emergence [35]). We transferred 50 newly emerged (<24 h old) and mated females from an abovementioned Petri dish onto a new leaf square (4 cm × 4 cm) on wet cotton wool in a new Petri dish. We prepared 230 such dishes and replaced all leaf squares once a day to obtain mated females of 1, 3, 6, 9, and 12 days old for the experiments. The lifespan of mated females of *T. ludeni* is approximately 15 days in the current experimental thermal conditions [36], and thus, we tested the dispersal of mated females up to 12 days old. To keep the number of females on each leaf square consistent prior to the experiments, we checked the leaf square every day and replaced dead females with live ones of the same age.

### 2.3. Dispersal and Reproduction of Females of Different Ages

We established a dispersal platform consisting of 21 leaf arenas (2 cm × 2 cm for the first leaf square and 2 cm × 1 cm for others) placed on a layer of wet cotton in a tray (45 cm length × 36 cm width × 1.5 cm heigh) (Figure 1). The primary leaf square was used to introduce experimental mites. Leaf arenas were connected linearly by Parafilm bridges (4.5 cm × 1.5 cm), with approximately 1 mm overlapping to reduce the influence of water between leaf arenas and bridges on mite dispersal. To minimize the possible interference of uneven light on mite dispersal, three parallel light tubes of around 80 cm in length with an interval of around 15 cm between two tubes were placed 100 cm above the experimental tray to ensure that all leaf arenas had similar light conditions, including the light intensity and direction. We performed 17, 14, 14, 14, and 16 replicates for the treatment of mites aged 1, 3, 6, 9, and 12 days old, respectively. All replicates were carried out under the same light condition as described above. For each replicate, we transferred 50 mated females of a desired age onto the first leaf square (primary leaf square) and allowed them to settle for 40 min. We then connected all leaf squares using Parafilm bridges (4.5 cm length × 1.5 cm width; Parafilm^®^, Waltham, MA, USA) and allowed mites to disperse freely. After 24 h, we removed all bridges, counted the number of adults on each leaf square, and then removed all adults and counted the number of eggs on each leaf square. All leaf squares with eggs were individually transferred onto a wet cotton pad in a Petri dish, and the eggs were reared on their original leaf squares until adulthood. We recorded the number of emerged adults and calculated the immature offspring (from egg to adult) survival rate.

We defined the dispersal probability as the percentage of mites that dispersed out of the primary leaf square. We numbered the leaf arenas progressively from the primary leaf square (number zero), to estimate the dispersal distance. We recorded the mean dispersal distance of a replicate by averaging the dispersal distance of all individuals in the replicate and determined the total number of eggs laid in a replicate by summing up the number of eggs on all leaf arenas. We recorded the mean distance of eggs of a replicate by averaging the distance of all eggs laid on different leaf arenas. 

### 2.4. Dispersal of Females at Different Densities

Because spider mite density in nature ranges from 0.1 to 50 individuals/cm^2^ [46,47], we set up four densities: 2.5, 12.5, 25.0, and 37.5 females/cm^2^, with 10, 50, 100, and 150 mated females on the primary leaf square, respectively. There were 15, 15, 16, and 16 replicates, respectively, for the above density treatments. For each replicate, we transferred 1-day-old mated females of a desired number according to the treatments on the primary leaf square and allowed them to settle for 40 min. We then connected all leaf arenas as above and allowed mites to disperse freely for 24 h, after which we removed all bridges and counted the number of adults on each leaf arena. We calculated the dispersal probability and distance of a replicate as described above.

### 2.5. Statistical Analysis

We analyzed all data using SAS software (SAS 9.4, SAS Institute Inc., Cary, NC, USA). Data on the dispersal probability in the age and density experiments, dispersal distance in the age experiment, total number of eggs laid, and number of eggs laid outside the primary leaf squares were normally distributed (Shapiro–Wilk test, UNIVARIATE Procedure), and thus analyzed using an ANOVA with Tukey’s test for multiple comparisons (GLM procedure). Data on the distance of eggs, number of eggs laid on the primary leaf square, and dispersal distance in the density experiment were square-root-transformed to achieve a normal distribution before the ANOVA. The survival of offspring on and outside the primary leaf squares was not normally distributed, even after transformation, and was analyzed using a non-parametric ANOVA (GLM procedure).

We used a logistic linear model, *y* = exp(*a* + *bx*), to determine the distribution of egg density (number of eggs/leaf square area) and changes in offspring survival rate over distance with a Gamma distribution and a log link function in all age treatments, where *y* is the response variable, *x* is distance, and *a* and *b* are the constant parameters of the model (GLIMMIX Procedure). 

## 3. Results

### 3.1. Dispersal and Reproduction of Females of Different Ages

Our results show that within the 24-h experimental period, 1-day-old females were significantly less likely to disperse compared to the older females (F_4, 70_ = 15.51, *p* < 0.0001) (Figure 2a), and they dispersed significantly shorter distance than older females (F_4, 70_ = 7.55, *p* < 0.0001) (Figure 2b). Furthermore, 1-day-old females laid significantly fewer eggs than older females, and 3-day-old females laid significantly more eggs than females of other ages (F_4, 70_ = 25.73, *p* < 0.0001) (Figure 2c).

Egg density significantly decreased over distance in all age treatments (Figure 3a; Appendix A). Females of different ages laid similar numbers of eggs on the primary leaf squares (F_4, 70_ = 1.62, *p* = 0.1797) (Figure 4a), but older females laid significantly more eggs outside the primary leaf squares than 1-day-old females (F_4, 70_ = 22.59, *p* < 0.0001) (Figure 4b). In addition, older females produced eggs over significantly longer distances than 1-day-old females (F_4, 70_ = 5.35, *p* = 0.0008) (Figure 4c). The immature survival rate significantly increased over distance in all treatments (Figure 3b; Appendix A). Offspring survival rates on or outside the primary leaf square were similar between females of different ages (F_4, 70_ = 1.16, *p* = 0.3339 for the primary leaf square; F_4, 69_ = 1.80, *p* = 0.1379 for outside the primary leaf square) (Figure 4d,e). 

### 3.2. Dispersal of Females at Different Densities

Mites were significantly more likely to disperse with an increase in population density (F_3, 58_ = 15.33, *p* < 0.0001) (Figure 5a). Similarly, they dispersed significantly longer distance within 24 h when the population density was higher (F_3, 58_ = 14.06, *p* < 0.0001) (Figure 5b).

## 4. Discussion

Our results show that compared to 1-day-old females of *T. ludeni*, older ones had significantly higher dispersal probabilities and longer dispersal distances and laid more eggs (Figure 2). These findings strongly suggest that age-dependent dispersal is caused by reproductive readiness when older females are capable of producing more mature eggs within 24 h [42,48]. In general, females with more eggs tend to disperse in wider areas [1]. We also show that 3-day-old females laid the highest number of eggs under our experimental setting (Figure 2c). Female spider mites usually have a short pre-oviposition period, followed by an increasing oviposition rate for the first few days of the oviposition period, and then reduce reproduction after reaching an oviposition peak [42]. Population density may change the daily egg-laying pattern such as the timing of oviposition peaks [42]. For example, in *T. ludeni*, mated females reach a reproductive peak after they are 7 days old, but females at higher densities reach a peak before 6 days old [42,49]. The 12.5 females/cm^2^ in this study represents a relatively high population density [46,47], and females laying the highest number of eggs at 3 days old may be attributed to a resource allocation strategy of *T. ludeni* to maximize reproduction during the early stage of their lifespan at high population densities [42]. In addition, we show that females had similar number of eggs after oviposition peaked at 3 days old (Figure 2c), suggesting that high density flattens the daily oviposition curve [42]. By comparing the dispersal and reproduction between 1-day-old females and older ones, we show synchronization of these two behaviors. Such a phenomenon has also been reported in many other insects, such as the light brown apple moth, *Epiphyas postvittana* (Walker) (Lepidoptera: Tortricidae) [14]; the tarnished plant bug, *Lygus lineolaris* (Palisot de Beauvois) (Hemiptera: Miridae) [25,50]; the rice bug, *Leptocorisa chinensis* Dallas (Hemiptera: Alydidae) [15]; and the butterfly, *Maculinea* (*Phengaris*) *teleius* Bergsträsser (Lepidoptera: Lycaenidae) [11]. For example, in *L. lineolaris*, females of pre-reproductive age fly much less than gravid older females, and females with higher egg loads are more likely to fly compared to those carrying few or no eggs [50].

The higher dispersal probability and longer dispersal distance of females ladened with more eggs may contribute to fast population growth at new sites, promoting pest outbreak and range expansion [22,24,50]. Furthermore, the similarly high dispersal probability and long dispersal distance among older females (3–12 days old) (Figure 2) suggest that *T. ludeni* females can quickly disperse and infest new host plants by walking for most of their adult lives. In their study on the dispersal behavior of *T. urticae*, Li and Margolies reveal that aerial dispersal is much more frequent in younger females because they are smaller in body size and may be more easily carried aloft by air currents [9]. With regard to ambulatory dispersal, Suiter and Gould show that adult females of different ages have a similar dispersal tendency in *T. urticae*; however, the purpose of their study was to test female dispersal in response to a pyrethroid insecticide that is known to stimulate dispersal behavior [12]. We provide the first evidence that females of ≥3 days old that are capable of producing more eggs within 24 h are more likely to disperse through walking. These results show the importance of early detection and management of pest mites on crops, such as at immature stages, to limit their distribution.

Dispersal serves multiple functions, such as reducing competition for resources between each other [51,52] and their offspring [1,11,17,18,19,20,53] and avoiding harsh conditions. We show that although older females had higher fecundity (Figure 2c), they laid similar number of eggs on primary leaf squares (Figure 4a) but produced significantly more eggs outside the primary leaf squares over longer distances compared to the 1-day-old ones (Figure 4b,c). These results suggest that older females of *T. ludeni* can reduce offspring competition for food resources by dispersing from their natal habitats to their new habitats (Figure 2a), and the spread of the local population and increase in the population size of spider mites within and between the adjacent habitats take place through the females with high reproductive output.

We show that with a decrease in egg density over the dispersal distance, offspring survival significantly increased (Figure 3), implying that crowding and competition negatively affect offspring fitness, and the rich resources available at their birthplace increase their fitness. This strategy is critical to offspring survival in spider mites because immatures have limited dispersal ability [9,45,54] and no parental care. A similar strategy has also been reported in the predatory mites *Phytoseiulus persimilis* Athias-Henriot and *Neoseiulus californicus* McGregor (Acari: Phytoseiidae). The mothers of these two species produce eggs on a patch according to the number of prey eggs on the patch that are sufficient for their offspring to survive/develop before dispersing away [55,56].

Previous studies show that species may exhibit positive or negative density-dependent dispersal, which depends on the balance of the costs and benefits of group living [3,9,27,28,29,30,31,32,57]. The present study indicates that *T. ludeni* females exhibited significantly increased dispersal probability and distance with an increase in population density (Figure 5). This could be attributed to the fact that high population density reduces individual and population fitness [3,42,43,44,58,59]. In *T. ludeni*, high population density negatively affected individual reproduction and population growth [42,43,44]. Our results suggest that females would avoid overcrowding conditions to reduce competition through dispersal over a longer distance. The positive density-dependent dispersal strategy has been reported in many animal species, such as the butterflies *Maculinea nausithous* and *M. teleius* (Lepidoptera: Lycaenidae) [19], the two-spotted spider mite *T. urticae* [9,30], and the brown garden snail *Cornu aspersum* (Müller) (Gastropoda: Helicidae) [60]. This strategy allows individuals to leave high-density habitats and emigrate to low-density habitats and/or colonize new habitats [3,6,8,61], promoting range expansion and the invasion speed of *T. ludeni*. Spider mites build up their populations rapidly and exhaust resources very fast; they establish a new population by dispersing away from crowded conditions and finding unexploited resources. Our results suggest that *T. ludeni* can travel relatively long distances (e.g., around 18 cm within 24 h at the highest density tested) and find new resources within and even between adjacent plants by crawling, especially between cultivated plants such greenhouse crops, which may facilitate pest outbreaks.

In this study, we tested dispersal using a linear experimental platform. This is a simple method which is widely used to test ambulatory dispersal in small arthropods like mites [27,30,36,62,63]. In some other studies, the study organisms are allowed to disperse in multiple directions. For example, in a study on *T. urticae*, experimental mites were introduced to the bottom of host plant leaves and were allowed to descend or ascend [2]. This setup allows researchers to observe within-plant mite distribution and is closer to field conditions. Further studies are warranted to test the dispersal strategies of *T. ludeni* in relation to age and density when given a choice of multiple dispersal directions or in field conditions.

In conclusion, increasing age and density promotes dispersal probability and distance in *T. ludeni*. Females that are capable of producing more mature eggs within 24 h are more likely to disperse and move longer distances. This dispersal behavior promotes oviposition away from their natal habitats and the site of the original arrival of the foundress female, consequently decreasing competition and increasing their own fitness and that of their offspring. These strategies may facilitate the pest outbreak and invasion success of *T. ludeni*.

## Figures and Tables

**Figure 1 insects-15-00387-f001:**
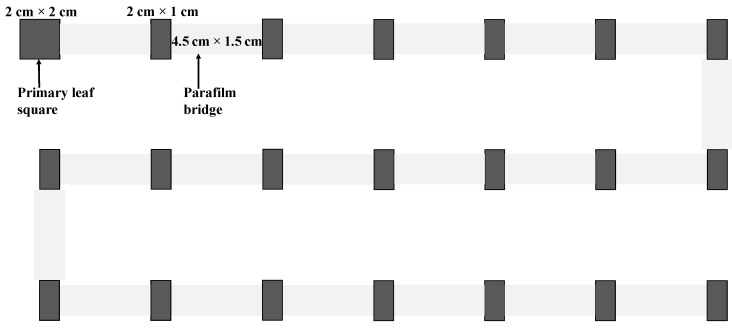
Dispersal platform for experiments, which consisted of 21 leaf arenas (2 cm × 2 cm for the first leaf square and 2 cm × 1 cm for others) placed on a layer of wet cotton in a tray (45 cm length × 36 cm width × 1.5 cm heigh). The primary leaf square was used to introduce the experimental mites. Leaf arenas were connected by Parafilm bridges (4.5 cm × 1.5 cm), with approximately 1 mm overlapping to reduce the influence of water between leaf arenas and bridges on mite dispersal.

**Figure 2 insects-15-00387-f002:**
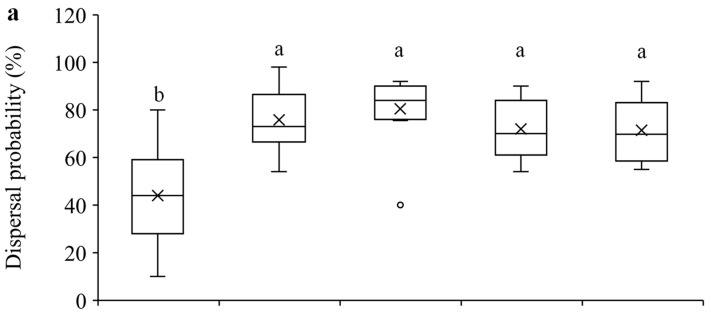
Mean dispersal probability (**a**), dispersal distance (**b**), and total number of eggs laid (**c**) within 24 h in *T. ludeni* females of different ages. Each box plot shows the maximum (⊤) and minimum (⊥) scores, the range of upper and lower quartiles (the box), and the mean (×) and median (line in the box) scores; the circle is an outlier of the minimum score. For each parameter, boxes with the same letters are not significantly different (*p* > 0.05).

**Figure 3 insects-15-00387-f003:**
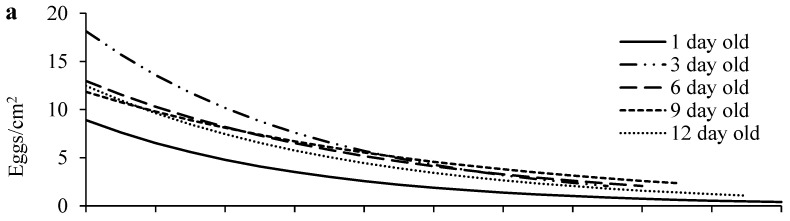
Egg density (**a**) and offspring survival (**b**) over distance in *T. ludeni* females of different ages. Raw data were subjected to analysis for all parameters.

**Figure 4 insects-15-00387-f004:**
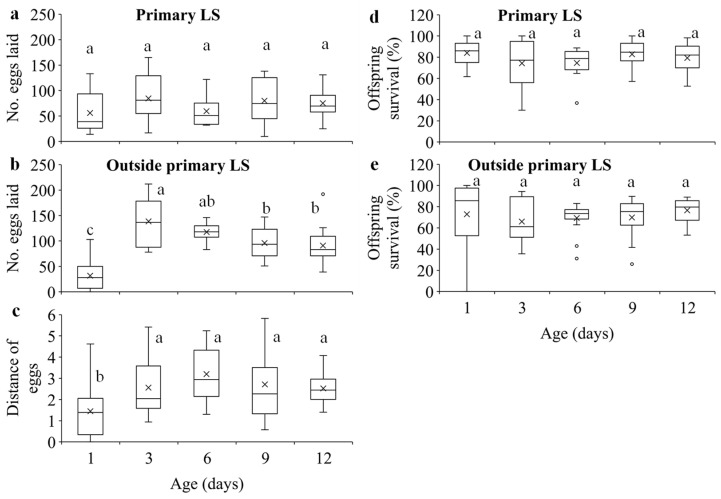
Number of eggs laid on (**a**) and outside (**b**) the primary leaf squares (LS), distance of eggs (**c**) within 24 h, and offspring survival on (**d**) and outside (**e**) the primary leaf squares in *T. ludeni* females of different ages. Each box plot shows the maximum (⊤) and minimum (⊥) scores, the range of upper and lower quartiles (the box), and the mean (×) and median (line in the box) scores; the circles are outliers of the minimum or maximum scores. For each parameter, boxes with the same letters are not significantly different (*p* > 0.05).

**Figure 5 insects-15-00387-f005:**
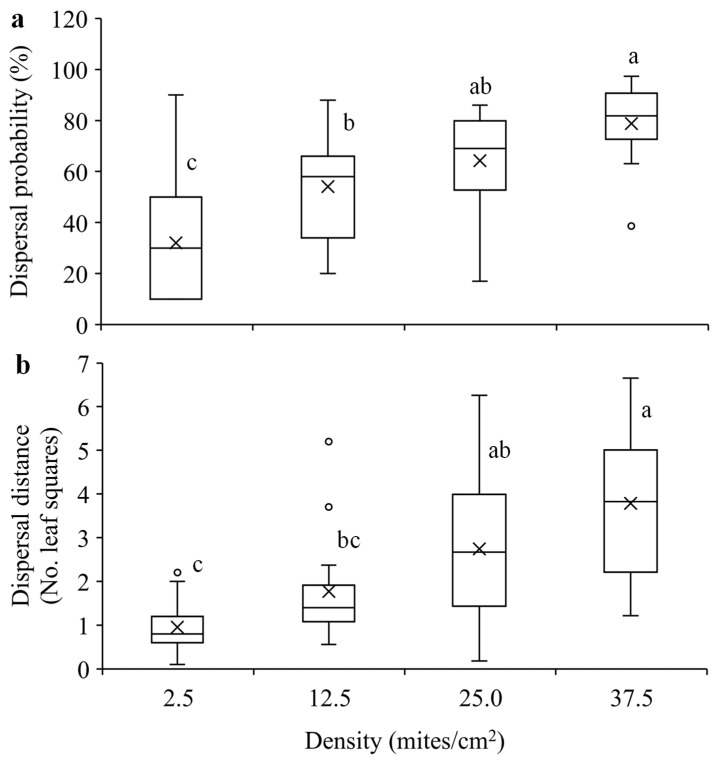
Mean dispersal probability (**a**) and dispersal distance (**b**) within 24 h in *T. ludeni* females of different densities. Each box plot shows the maximum (⊤) and minimum (⊥) scores, the range of upper and lower quartiles (the box), and the mean (×) and median (line in the box) scores; the circles are outliers of the minimum or maximum scores. For each parameter, boxes with the same letters are not significantly different (*p* > 0.05).

## Data Availability

The raw data supporting the conclusions of this article will be made available by the authors on request.

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
