# Peer review of "Age and Density of Mated Females Affect Dispersal Strategies in Spider Mite Tetranychus ludeni Zacher"

_insects, 2024, doi:10.3390/insects15060387_

Round 1

Reviewer 1 Report

Comments and Suggestions for Authors

Dear authors,

I wish to congratulate you on an excellent article, revealing the strategies of this spider mite species to survive. 

1. The survival dynamics of a spider mite pest.

2. The influence of the age of fertilised females on the oviposition rate and implication on survival of offspring.

3. This helps to implement more effective control strategies.

4. As it is, the methodology of this study was very complex and depended on effective teamwork because of the many replicates and its maintenance. Search for new technologies, if available, are the only controls I can suggest.

5. The references are appropriate.

6. This is a straightforward study, and the figures and tables add the necessary quality.

Author Response

Reviewer 1:

Dear authors,

I wish to congratulate you on an excellent article, revealing the strategies of this spider mite species to survive.

  1. The survival dynamics of a spider mite pest.
  2. The influence of the age of fertilised females on the oviposition rate and implication on survival of offspring.
  3. This helps to implement more effective control strategies.
  4. As it is, the methodology of this study was very complex and depended on effective teamwork because of the many replicates and its maintenance. Search for new technologies, if available, are the only controls I can suggest.
  5. The references are appropriate.
  6. This is a straightforward study, and the figures and tables add the necessary quality.

Our response: We appreciate your highly positive comments on our manuscript.

Reviewer 2 Report

Comments and Suggestions for Authors

The text is adequately written. I have some questions about the methodology, which I hope the authors could explain, regarding:

a) the experimental setup used in the work - a rectangular tray, releasing the mites at one angle, instead of releasing the mites in the center of an area containing rearing units distributed in concentric circles of different rays.

b) the possible interference of the position of the light source(s) in relation to the position of the experimental trays.

I have presented some suggestions to be considered by the authors, especially in relation to:

a) emphasize in the manuscript that it deals with ambulatory dispersal.

b) the inclusion (in Introdution or Discussion) of addition papers.

c) a slightly more informative section concerning the number of "mature" eggs held by each female at a given time.

d) relate the observed oviposition dynamics with the pattern usually expressed by tetranychids in general, even when isolated into an experimental unit, without the possibility for dispersal (see ref. 32 of this manuscript). I think this is quite important.

e) I do not consider myself sufficiently qualified to talk about the appropriateness of the analysis adopted for comparison of dispesal distances, considering that (as I understand) those numbers are not continuous. I suggest confirming that with a statistician.

Other comments in the attached pdf.

Comments on the Quality of English Language

As I understand, it is quite well written. I added a few suggestions in this regard.

Author Response

Reviewer 2: The text is adequately written. I have some questions about the methodology, which I hope the authors could explain, regarding:

  1. a) the experimental setup used in the work - a rectangular tray, releasing the mites at one angle, instead of releasing the mites in the center of an area containing rearing units distributed in concentric circles of different rays.

Our response: Thanks for your comment. We have discussed this briefly in Discussion. If using a circular arrangement, there would be other factors that must be taken into consideration, e.g., the dispersal direction and the number of mites at each direction, making the statistical analysis and explanation very complicated, essentially impossible for the data. By contrast, linear dispersal arena is a simple way to test ambulatory dispersal.

Reviewer 2: b) the possible interference of the position of the light source(s) in relation to the position of the experimental trays.

Our response: The lights are positioned just above the experimental trays to minimize the possible interference of uneven lights on mite dispersal, i.e., all leaf arenas in a replicate had similar light condition including the light intensity and direction and all replicates were carried out under the same light conditions. We have added such information in Materials and Methods.

Reviewer 2: I have presented some suggestions to be considered by the authors, especially in relation to:

  1. a) emphasize in the manuscript that it deals with ambulatory dispersal.
  2. b) the inclusion (in Introdution or Discussion) of addition papers.
  3. c) a slightly more informative section concerning the number of "mature" eggs held by each female at a given time.
  4. d) relate the observed oviposition dynamics with the pattern usually expressed by tetranychids in general, even when isolated into an experimental unit, without the possibility for dispersal (see ref. 32 of this manuscript). I think this is quite important.
  5. e) I do not consider myself sufficiently qualified to talk about the appropriateness of the analysis adopted for comparison of dispesal distances, considering that (as I understand) those numbers are not continuous. I suggest confirming that with a statistician.

Other comments in the attached pdf.

Our response: Thanks for your suggestions. The points raised here are the same with those in the attached PDF file. We have addressed them point by point, please see the detailed responses below.

Reviewer 2: Comments on the Quality of English Language

As I understand, it is quite well written. I added a few suggestions in this regard.

Our response: Thanks for your positive comment.

Reviewer 2: L1. Perhaps interesting to include in the Discussion:

1) Zhou et al. (2021) DOI: 10.1093/jee/toaa313

  1. s) Zhou et al. (2022) Zoosymposia, 22: 122 - Abstract of the present manuscript, apparently.

1) Scahusberger et al. (2019) DOI: 10.1098/RSOS.191201

2) Schausberger et al. (2021) DOI: 10.3389/fevo.2021.745036

Our response: Thanks. We have added three of the four suggested references. Scahusberger et al. (2019) deals with group living and mating between different phenotypes of individuals and alternative reproductive tactics, which is not much related to this manuscript, and thus excluded in Discussion.

Reviewer 2: L10. The "Simple Summary" is basically the same as the "Abstract". It seems to conveninet for the authors to make the "Simple Summary" more compact.

Our response: Thanks. The content meets the requirement of the Simple Summary although it is similar to the Abstract. The purpose of the Simple Summary is to describe the work simply and concisely to the public with no technical terms, which contains a clear statement of the problem addressed, the aims and objectives, pertinent results, conclusions and how they will be valuable to society.

Reviewer 2: L15-17. Add “of age ande density” and delete “and that of density on dispersal probability and distance.”

Our response: Thanks. We have revised accordingly.

Reviewer 2: L17. “older females with more eggs” Please see my suggestions in this regard. I should add that although I have been working on biology and taxonomy of tetranychids for many years, I do not remember seeing eggs inside the females (differently from what I see in Mesostigmata and some Astigmata). I do not know why. Thus, I suggest the authors to consider this, and provide more information in the paper in this regard. I say that because it does not seem adequate just to indicate that "older females have more eggs". Based on illustrations in the literature, I suppose the tetranychid females have in place a series of devolping eggs at different maturity stages, given that part of the can produce several eggs (some over 20, as T. evansi) a day. But I do not expect to have more than one totally mature egg at one time, given that as far as I know each female lays one egg and then (most often) only after a few hours will lay the next egg. This aspect may be adequately mentioned by the authors of this publication somewhere (perhaps in more detail in Discussion)? I have seen from the literature that at least some of the authors have published several papers on dispersal and reproduction of T. ludeni. Sorry for being so lengthy!

Our response: Thanks for your patient explanations. We have revised the descriptions accordingly.

Reviewer 2: L21 and L33. “to avoid crowding” In my opinion, that is not done FOR A PURPOSE. It is done naturally and that favors the species!

Our response: Thanks. It is well accepted that crowding leads to competition which may drive the dispersal of individuals, i.e., crowding is one of the reasons of individual dispersal. Therefore, we feel that it is appropriate to say that mites dispersed “to avoid crowding”.

Reviewer 2: L45. I suggest deleting “To date”.

Our response: Thanks. We have deleted it.

Reviewer 2: L51. “aiming to increase”. Again, in my oppinion, it just happens, and by favoring the species it is selected for.

Our response: Thanks. We have deleted “aiming to” and revised the descriptions.

Reviewer 2: L51. Same comment. Please consider the following option:

"The spreading of offspring over habitats of high quality may be particular important for species whose ....

Our response: Thanks. We have revised accordingly.

Reviewer 2: L63. “as reducing predation risk and increasing foraging efficiency”. For spider mites, high density may actually favor Phytoseiulus predators (one of the main groups of predatory mites of tetranychid prey). Additionally, it seems that forage would not be more efficient at high densities of tetranychids. Please consider changing this sentence.

Our response: We did not limit the phenomenon to spider mites. But we have made it clearer.

Reviewer 2: L69. I sugget "whose" instead of "where"

Our response: Thanks. We have revised accordingly.

Reviewer 2: L70. Change “host” into “host species”

Our response: Thanks. We have revised accordingly.

Reviewer 2: L82-83. “So far, it is unknown whether age-specific ambulatory dispersal is related to reproduction” Sorry, this does not seem sufficently clear.

Our response: Thanks. We have revised to make it clearer.

Reviewer 2: L84-86. “In this study, we used T. ludeni to investigate the effect of age on dispersal probability and distance and subsequent reproduction and that of density on dispersal probability and distance.” Please see my suggestion above. In addition, it seems to me that it would be important to emphasize that this work deals with short-range dispersal (ambulatory dispersal).

Our response: Thanks. We have added this information.

Reviewer 2: L91. Everywhere in the text, I sugges associating "distance" with another term that would make the reasoning more understandable (perhaps "daily travelled distance" or something considered more appropriate by the authors.

Our response: We have defined that mites were allowed to disperse for 24 hours and recorded their dispersal distance. We believe that the "distance" used in the text is clear.

Reviewer 2: L109. “prepared below” with the characteristics indicated below.

Our response: Thanks. We have revised this accordingly.

Reviewer 2: L114-115. with the characteristics indicated in the previous paragraph to each of those Petri dishes

Our response: Thanks. We have revised this accordingly.

Reviewer 2: L123. Two points:

  1. a) Any care in relation to the possible effect of the position of each tray in relation to light source? It seems important to discuss mention something in this regard.
  2. b) In some papers in the literaure, the setup includes the distbution of plants (or plant parts) in concentric circles of different rays around a mite source. What are the advantages or disadvantages of the setup used here? Perhaps this could be quickly touched upon (not here, but in Discussion).

Our response: a) The lights are positioned just above the experimental trays to minimize the possible interference of uneven lights on mite dispersal, i.e., all leaf arenas in a replicate had similar light condition including the light intensity and direction and all replicates were carried out under the same light condition. We have added such information in Materials and Methods.

  1. b) We have briefly discussed the dispersal setup in discussion.

Reviewer 2: L124. In these cases, these are rectangles, not squares. Thus, my suggestion is to refer to them as "leaf arenas" of "leaf experimental units" (suitable for both, squares and rectangles).

Our response: Thanks. We have revised it accordingly here and in some other parts of the manuscript.

Reviewer 2: L125. Knowing that light affects migration, how was the position of the light sources accounted for in the experiment?

Our response: The lights are positioned just above the experimental trays to minimize the possible interference of uneven lights on mite dispersal, i.e., all leaf arenas in a replicate had similar light condition including the light intensity and direction and all replicates were carried out under the same light conditions. We have added such information in Materials and Methods.

Reviewer: L125. on a layer of wet cotton.

Our response: Thanks. We have revised accordingly.

Reviewer 2: L130-132. freely. After 24 h, we removed all bridges and adult mites, after counting the numbers of adults and eggs in each arena.

Our response: Thanks. Actually, we removed the adults first and then counted the eggs, which avoided females laying additional eggs between the interval of counting the number of adults and eggs. We have revised the sentence to make it shorter and simpler.

Reviewer 2: L134. “until adulthood” Why? It seems important to say. Did the author sexed the mites? Could this infomation be introduced into the manuscript? Any differences in sex ratio detected for females of differen ages?

Our response: We counted the number of emerged adults to see how offspring survival would be influenced by competition. Data regarding offspring sex ratio are not closely related to the topic of this manuscript, and thus we did not include them into the manuscript.

Reviewer 2: L136. We numbered the leaf arenas progressively from the primary leaf square (number zero), to estimated dispersal distance.

Our response: Thanks. We have revised accordingly.

Reviewer 2: L140. “leaf squares” leaf arenas of the replicate.

Our response: Thanks. We have revised accordingly.

Reviewer 2: L161-164. I do not consider myself sufficiently qualified to talk about the appropriateness of this type of analysis, considering that those numbers are not continuous.

Our response: The dispersal distance and number of eggs are discrete variables, and it is appropriate to use ANOVA for analysis.

Reviewer 2: L224. “in T. ludeni” I sugges deleting.

Our response: We have deleted it.

Reviewer 2: L226. “when older females carry more mature eggs” I am not sure this is the most adequate way of expressing the process. This might suggest that a tetranychid would (like some Astigmata, for example) carry around well formed eggs inside. In tetranychids , is that instead, more eggs become mature at each day. I suggest the authors give a little more detail in this regard, based on information available in the literature. In my view, that would add to quality of the manuscript.

Our response: Thanks for your suggestions, however, we feel that the relationship of egg formation and egg laying behaviour is beyond the scope of our study. Discussing that would be inappropriate. We have revised and changed the expression into “are capable of producing more mature eggs”.

Reviewer 2: L231-236. Please take into account the dynamics of oviposition of the tetranychids (as shown in ref. 42), with a short pre-oviposition period, followed by a growing oviposition rate for the first few days of the oviposition period, up to  maxium, with oviposition reducing afterward, usually slowly, to reach a rather short post-oviposition (no oviposition) phase. That somewhat coincides with the pattern shown in the figures provided in the manuscript.

Our response: Thanks. We have revised and discussed this.

Reviewer 2: L241. “infest new host plants” While it is right that dispersal between neighboring plants by walking (as in this experiment), the discussion should take into account the much higher complexity of the process in the field, where dispersal at longer distance would most often involve mites carried by the wind, as mentioned in the following sentence. I suggest the authors to stress the fact that in this study short (not long) range dispersal is under evaluation.

Our response: Thanks. We have revised the sentence.

Reviewer 2: L245. “carrying more eggs” Please see my previous comment. Also, please take into account the usual oviposition dynamics in "non-dispersing" usual development.

Our response: We have revised the expression.

Reviewer 2: L246-247. “as avoiding harsh conditions and reducing competition for resources between their own [50,51] and their offspring” I suggest inverting these 2 aspects (talk abut avoiding competittion first).

Our response: Thanks. We have revised accordingly.

Reviewer 2: L248. “older females” Not actually "older", but three day old females. Please notice a tendency for reduced oviposition, which reflects quite well the usual oviposition dynamics of tetranychid species (as shown in many papers).

Our response: Although females generally reduce oviposition after their reproductive peaks, the statement in the text is based on the results generated. The “higher” is compared between 1-day-old females and 3- to 12-day-old females.

Reviewer 2: L257. (Figure 4) Not Figure 3b??

Our response: Thanks for pointing it out. We have checked and revised throughout the paper accordingly.

Reviewer 2: L269-270. See previous comments: in my opinion, the female does avoids overcrowding and that in turn is favorable for the species (instead of saying that females avoids overcrowding for that purpose).

Our response: Thanks. It is well accepted that crowding leads to competition which may drive the dispersal of individuals, i.e., crowding is one of the reasons of individual dispersal. Therefore, we believe that it is appropriate to say that mites dispersed “to avoid crowding”.

Reviewer 2: L274. “This strategy” It seems better to me saying "This behaviour" instead of "This strategy".

Our response: We believe that strategy is appropriate to use here, for example, the phrase “density dependent dispersal strategies” is widely used in literature.

Reviewer 2: L278. “more mature eggs” Actually this fact was not investigated in this work. Thus, if the authors imply this really happens, then references showing that  should be provided.

Our response: We have revised our expression.

Reviewer 2: L279-280. “These dispersal strategies spread eggs out of their natal habitats and over longer distances, which reduces” Suggestion: The dispersal behavior promotes the oviposition away from the site of original arrival of the foundress female, consequently decreasing .... and increasing ....

Our response: Thanks. We have revised this accordingly.

Reviewer 3 Report

Comments and Suggestions for Authors

Review of the manuscript “Age and Density of mated females affect dispersal strategies in a spider mite”

            The present manuscript studied the age and density-dependent dispersal of European native spider mite, Tetranychus ludeni Zacher, on square leaf sections of bean plants. The manuscript is well written where methodology is well described for each step. The results are precise and straight to the point. The figures are well defined too. However, there are some critical questions, which must be addressed. I suggest for Major Revisions based on following queries:

1. The need of research statement requires justification.

The experiment-based research usually requires a hypothesis, derived from keen observations. This proposes the question (research problem) to be answered through experimentations. I was not able to find a proper research problem that authors wanted to solve. For instance, page 2 lines 45-46 and 76-77 only state the information unknown. An aspect unknown does not warrant the need of research, rather than its "need" and potential use to understand ecological and behavioral phenomenon. It is unclear why authors would like to study the age specific density dependent dispersal of T. ludeni.

            The literature review in such field of research shows that it is clearly known for spider mites that the mated females disperse under the effect of over-crowding and resource depletion. It is also known that older females can regulate sex ratio of the developing population and can also regulate the size and structure of the population. It has been studied previously that females are foundresses means that they will disperse only, while males and immatures do not. Additionally in recent years (see references 42, 43, 44) there is much similar work done already.

            I suggest authors to re-evaluate their need of research statement. One aspect would be useful where the subsequent off-spring survival and fecundity could be studied for T. ludeni but again the “WHY?” must be clear through objectives.

2. The methodology to conduct age and density dependent experiments should be reconsidered.

            I would like to appreciate authors for writing the first part of the methodology (preparation of mated females) in very clear and precise details. But some questions related to overall all experiments still remain. On page 3, line 110, those selected 50 randomly selected females were of what age, is not clearly mentioned. In line 114, the authors introduced males to quiescent deutonymphal females. How the gender could be known at quiescent deutonymphal stage? The authors gave 24 hours for whole of the study. Are these 24 hours sufficient? Additionally, was there any interval for data recording? In my opinion, it is crucial to make early observations and with intervals over the time period to clearly understand the factors involved in dispersal. One page 4, line 143, the authors referenced to the information that “Because spider mite density in nature ranges from 0.1 to 50 individuals/cm2”. This density is at what level of infestation is not clear. Also, the leaf section authors used for their experiments is not uniform (4 cm2 and 2 cm2 the rest). It could cause another issue. Additionally, the experimental layout is linear. I wonder why authors decided to study the dispersal behavior in one direction where mites have direction to move. Rather such experiments should be conducted in free environments where mites have multiple exits and entries.

3. The discussion of the manuscript should be re-written.

Although the results and written precisely, the discussion appears to be written hastily. The authors are comparing their results to the findings of other insects without considering their biological and ecological aspects. The authors in their experiments provided finite environment to mites, this could not be explained under the impression of “resource allocation” as mentioned by authors. It would be less likely to happen. Furthermore, the ballooning was not studied in the present study as the method for dispersal hence, it can not be discussed.

For other comments and suggestions, please see the attached PDF file.

Comments on the Quality of English Language

N/A

Author Response

Reviewer 3: The present manuscript studied the age and density-dependent dispersal of European native spider mite, Tetranychus ludeni Zacher, on square leaf sections of bean plants. The manuscript is well written where methodology is well described for each step. The results are precise and straight to the point. The figures are well defined too. However, there are some critical questions, which must be addressed. I suggest for Major Revisions based on following queries:

Our response: We appreciate the reviewer’s positive comments.

Reviewer 3: 1. The need of research statement requires justification.

The experiment-based research usually requires a hypothesis, derived from keen observations. This proposes the question (research problem) to be answered through experimentations. I was not able to find a proper research problem that authors wanted to solve. For instance, page 2 lines 45-46 and 76-77 only state the information unknown. An aspect unknown does not warrant the need of research, rather than its "need" and potential use to understand ecological and behavioral phenomenon. It is unclear why authors would like to study the age specific density dependent dispersal of T. ludeni.

Our response: We agree that the need of research should aim to understand ecological and behavioral phenomenon. We introduced research problems or gaps a few times in introduction, for example, at the end of first (L45-46) and fourth (L82-83) paragraph of introduction. We introduced factors that may affect age-specific dispersal in the second paragraph, and this research aims to test whether age-specific dispersal behaviour is related to age-specific reproduction. Based on the negative reproductive performance of T. ludeni at high densities, we hypothesized that high density may promote dispersal probability and distance. Results generated would facilitate our understanding of the dispersal strategies of T. ludeni and whether factors affecting dispersal probability would also affect dispersal distance.

Reviewer 3: The literature review in such field of research shows that it is clearly known for spider mites that the mated females disperse under the effect of over-crowding and resource depletion. It is also known that older females can regulate sex ratio of the developing population and can also regulate the size and structure of the population. It has been studied previously that females are foundresses means that they will disperse only, while males and immatures do not. Additionally in recent years (see references 42, 43, 44) there is much similar work done already.

I suggest authors to re-evaluate their need of research statement. One aspect would be useful where the subsequent off-spring survival and fecundity could be studied for T. ludeni but again the “WHY?” must be clear through objectives.

Our response: Yes, we mentioned that in general female spider mites disperse under over-crowding and resource depletion conditions, but how different population densities would affect dispersal probability and distance is largely unknown in spider mites. The references 42, 43, 44 investigated the effect of population density and size on reproductive performance of T. ludeni, but it is still not clear the dispersal strategies of the species in response to different population densities. Therefore, the study topic of those references and the present study are different.

The fecundity and subsequent off-spring survival after dispersal are part of the dispersal strategies we aim to investigate. It helps to explain the different dispersal probability and distance of females of different ages.

Reviewer 3: 2. The methodology to conduct age and density dependent experiments should be reconsidered.

I would like to appreciate authors for writing the first part of the methodology (preparation of mated females) in very clear and precise details. But some questions related to overall all experiments still remain. On page 3, line 110, those selected 50 randomly selected females were of what age, is not clearly mentioned. In line 114, the authors introduced males to quiescent deutonymphal females. How the gender could be known at quiescent deutonymphal stage? The authors gave 24 hours for whole of the study. Are these 24 hours sufficient? Additionally, was there any interval for data recording? In my opinion, it is crucial to make early observations and with intervals over the time period to clearly understand the factors involved in dispersal. One page 4, line 143, the authors referenced to the information that “Because spider mite density in nature ranges from 0.1 to 50 individuals/cm2”. This density is at what level of infestation is not clear. Also, the leaf section authors used for their experiments is not uniform (4 cm2 and 2 cm2 the rest). It could cause another issue. Additionally, the experimental layout is linear. I wonder why authors decided to study the dispersal behavior in one direction where mites have direction to move. Rather such experiments should be conducted in free environments where mites have multiple exits and entries.

Our response: Thanks for your detailed questions and suggestions. The questions raised here are the same as those in the attached PDF file. We have explained your questions point by point. Please see the detailed responses below.

Reviewer 3: “Also, the leaf section authors used for their experiments is not uniform (4 cm2 and 2 cm2 the rest).”

Our response: Similar with the question of one direction dispersal arena, we believe the different size of leaf squares within a replicate would not affect the testing of age and density effect as long as all the treatments are using the same experimental setup.

Reviewer 3: 3. The discussion of the manuscript should be re-written.

Although the results and written precisely, the discussion appears to be written hastily. The authors are comparing their results to the findings of other insects without considering their biological and ecological aspects. The authors in their experiments provided finite environment to mites, this could not be explained under the impression of “resource allocation” as mentioned by authors. It would be less likely to happen. Furthermore, the ballooning was not studied in the present study as the method for dispersal hence, it can not be discussed.

Our response: Thanks for your comments. The comments here are the same as those in the attached PDF file. We have explained them point by point below.

Reviewer 3: “Furthermore, the ballooning was not studied in the present study as the method for dispersal hence, it can not be discussed.”

Our response: We mentioned this not to discuss ballooning dispersal. The purpose is to present the distinction between previous and current studies.

Reviewer 3: For other comments and suggestions, please see the attached PDF file.

Our response: Thanks. We addressed your comments and suggestions point by point as follows.

Comments in PDF manuscript:

Reviewer 3: L2-3. “Title” As the keywords do not have the species name under study, I suggest to mention that in here. Like this, the current title represents a review paper rather than a research paper.

Our response: Thanks. We added the species name to the Title.

Reviewer 3: L12. “Simple Summary:” I suggest authors to differentiate between abstract and simple summary. Both here contain almost the similar text.

Our response: Thanks. We believe that the content meets the requirement of the Simple Summary although it is similar to the Abstract. The purpose of the Simple Summary is to describe the work simply and concisely to the public with no technical terms, which contains a clear statement of the problem addressed, the aims and objectives, pertinent results, conclusions and how they will be valuable to society.

Reviewer 3: L36. “Keywords” I suggest to rearrange alphabetically.

Our response: Thanks. We have rearranged them.

Reviewer 3: L39. “Introduction” Strangly, after reading the first 2 paragraphs of introduction, it remains unclear that authors referring to what? an insect, a mite, general arthropods, other organisms etc. I suggest to rephrase them and do not conclude here what is poorly studied area? generally, the flow of introduction gives the evidence first and then concludes what should be studied.

Our response: We appreciate your suggestion on introduction writing. However, we believe that there are many ways for writing introduction. In our manuscript, the first three paragraphs introduced the topic of this study, i.e., dispersal probability and distance and the factors that may affect them. We then introduced the study animal T. ludeni to investigate how the factors will affect dispersal probability and distance in this specific species.

Reviewer 3: L61. “Increasing density would raise dispersal probability”

Its not always like that. As long as the quality of resource (here food) is not declining, the organism (eg. spider mites) do not opt for dispersal rather than go for some structural changes in the existing population.

Similarly in other insect pests, the dispersal is an exhaustive phenomenon and usually preferred as last option.

Our response: We agree that increased density does not always raise dispersal probability, and we used “would” to stress this issue and changed to “would potentially”.

Reviewer 3: L62-63. “However, individuals living in high density may also have benefits such as reducing predation risk and increasing foraging efficiency ” The reference cited is for spiders, not for spider mites or insects. Hence, it became very irrelevant here. Plus, it is difficult to understand the purpose of this statement here.

Our response: Here we reviewed the topic on how different factors affecting species dispersal strategies. Therefore, the references are not limited to spider mites.

Reviewer 3: L64. Although could be true, but totally a contrasting stating to what was stated above. I suggest to rephrase the whole paragraph or justify the need of such statements.

Our response: The contrasting statements aim to raise a phenomenon that species have different characteristics/reasons that may lead to different dispersal strategies in response to density, which needs further species-specific investigation. The paper will investigate/analyze the characteristics and dispersal strategy of T. ludeni.

Reviewer 3: L65. As it is not the review paper, I suggest authors to be specific here what animal that would be.

Our response: As we mentioned in previous response, this paragraph reviewed the topic of this study and thus the contents are not limited to specific species.

Reviewer 3: L73-74. As this is known generally for spider mites, then why it should be studied for T. ludeni?

Our response: Although dispersal is common in crowding conditions in other spider mites, we cannot assume that it is the same in T. ludeni. Furthermore, it is not clear how different density would affect dispersal probability and dispersal distance.

Reviewer 3: L77-83. An aspect unknown does not warrant the need of research, rather than its "need" and potential use to understand ecological and behavioral phenomenon. It is unclear why authors would like to study the age specific density dependent dispersal of T. ludeni.

Our response: We agree that the need of research should aim to understand ecological and behavioral phenomenon. We addressed the factors that may affect age-specific dispersal in the second paragraph, and information provided here leaded to one of our research aims, i.e., to test whether age-specific reproduction is related to age-specific dispersal.

Reviewer 3: L84. “we used” it would not be a suitable word here.

Our response: Thanks. We rephrased the sentence.

Reviewer 3: L84. why are you using T. ludeni?

Our response: T. ludeni is an invasive species which has now invaded all continents except Antarctica. Dispersal is an important aspect of species’ invasion success. Therefore, we investigate dispersal strategies in relation to different factors using T. ludeni as a model species.

Reviewer 3: L86-90. It is not suitable here. Its better to be in Methdology.

Our response: Here, we previewed the experimental setup in relation to age and density then stated the hypotheses of this study, which will improve the continuity of content. Details are given in M&M.

Reviewer 3: L90. What can define that?

Our response: Thanks for pointing it out. We have clarified this.

Reviewer 3: L110. “We randomly selected 50 adult females from the colony” These females are of random age? How it was ensured that they were mated or unmated?

Our response: Yes, these females are of random ages. These randomly selected females were not used directly for experiments but used for producing experimental mites (female offspring) only. Therefore, it was not necessary to ensure that they are mated or unmated at this step as long as they can produce sufficient number of mites for experiments. Detailed information about experimental mites is given in M&M.

Reviewer 3: L114. how it was distinguished that those were female quiescent? Gender is difficult assess in deutonymphs.

Our response: Sex can be determined at deutonymph stage. Female quiescent deutonymphs are bigger than male quiescent deutonymphs. At the deutonymph stage, the hysterosoma of the female is markedly enlarged while that of the male is tapered towards the anal region.

Reviewer 3: L131-134. 24 hours would be enough to measure the dispersal and laid eggs?Under what effect this dispersal was evaluated? Host quality or density? As the heading states the effect of different age, but we established before that unless there is competition, the dispersal of spider mites may not occur.

Our response: Because we tested the effect of age on dispersal, longer dispersal period would be not appropriate to define female ages. We stressed in the introduction that high density did cause competition and reduce individual reproduction and population growth in T. ludeni, and 12.5 females/cm2 in this text is higher than that in the cited reference.

Reviewer 3: L138-139. I would like to know why authors decided to study the dispersal in linear direction. It means, the mites will have only 1 direction to go, so the real effect of a factor could not be known. It would be good if mites are given a choice like a circular arrangement of treatments my help.

Our response: Because all treatments used the same linear-direction dispersal arena, the effect of age or density are the only factor affecting dispersal. If we used a circular arrangement, there would be other factors that must be taken into consideration, e.g., the dispersal direction and the number of mites at each direction, making the statistical analysis and explanation complicated. By contrast, linear dispersal arena is a simple way to test ambulatory dispersal. We have briefly discussed different dispersal setups in discussion.

Reviewer 3: L143. “Because spider mite density in nature ranges from 0.1 to 50 individuals/cm2” this is at what time of infestation? it is very important to know.

Our response: These are field investigations of mite density, which are used as a reference for our density treatments. In nature, the population density increases over time depending on the environmental conditions. It is difficult to identify what time of infestation.

Reviewer 3: L185-188. As I understood that the experiment continued for 24 hours, it is unclear in the methodology that how the experiment continued for the immature data collection.

Our response: We stated this in L133-134. We have revised to make it clearer.

Reviewer 3: L227-229. “We also show that 3-d-old females laid the highest number of eggs under our experimental setting (Figure 2c) where the population density is relatively high”. I could not understand this. In the age related experiment, the density was fixed at 50 individuals, then how this statement could be justified?

Our response: We said that “where the population density is relatively high” because the fixed 50 individuals on a 4 cm2 leaf square is relatively high compared with the density in nature and density treatment in the cited references. We did not mean to compare density between different age treatments here. We have revised the sentence to make it clearer.

Reviewer 3: L229-231. Well it should not be confused with reproductive readiness. It is strange that spider mites would go for resource allocation at high density level in a finite environment. It could trigger dispersal. I suggest authors to verify the statement here.

Our response: Thanks. Readiness means that females are capable of reproducing eggs as they age while resource allocation strategy means that reproductive females may adjust reproduction according to social environment. We have added an example and cited two relevant references to verify the statement.

Reviewer 3: L231. what kind of synchronization was that? What kind of biology those insects had? was that similar to the mite species under study? what kind experimental arena was used in those studies? It would be not true, to generally quote examples without understanding the underlying concepts.

Our response: The synchronization refers to high dispersal tendency at the time of high reproductive output across species. This synchronization may have similar reasons and outcomes, e.g., lead to fast population growth when arrive at new sites regardless of whether they are similar species.

Reviewer 3: L244-245. I would not suggest to claim that here. Primarily because the experimental setup did not allow the ballooning or any other behavior for dispersal. Hence, the mites had one direction to move.

Our response: Yes, we did not test mite ballooning, and thus we restricted in the sentence that our results is about dispersal through walking. As explained earlier, one direction for dispersal is appropriate to test the effect of age and density on ambulatory dispersal.

Reviewer 3: L272-273. please mention like previous species.

Our response: Thanks. We have revised it accordingly.

Reviewer 4 Report

Comments and Suggestions for Authors

This is a well thought out experiment, well presented manuscript. The results and conclusions are clear. I only have some minor suggestions. In the discussion or introduction, it would be important to include the reported lifespan of the mites. Given that the experiment looked at fecundity and dispersal up to 12 days, having the expected lifespan would five the results more perspective. For example, if this mite species typically live 20 days vs 60 days, it would make a difference.

The authors discuss the implications of the result in terms of pest behavior and outrbreaks. Can there be some implications for management? For example, would early detection be important given that expansion increases with age?

One final point. The authors talk about range expansion. Their results show that the mites increase their dispersal distance as they age and as their density increases. For the highest density tested, the average distance dispersal was close to 4. In this study, that was close to 18 cm. In the field, it is difficult to measure this distance, but how does 18 cm relate to field conditions? Would this mostly allow within or between plant movement? It is understandable that the number of leaf squares is used as a relative distance, but I feel trying to relate that to field conditions would be appropriate.

Fig. 3 - can you provide the r-squares for the curves

249: Fig. 4a

251: Fig. 4b-c

257: fig. 3

Comments on the Quality of English Language

English is good, I only detected minor grammar issues.

Author Response

Reviewer 4: This is a well thought out experiment, well presented manuscript. The results and conclusions are clear. I only have some minor suggestions. In the discussion or introduction, it would be important to include the reported lifespan of the mites. Given that the experiment looked at fecundity and dispersal up to 12 days, having the expected lifespan would five the results more perspective. For example, if this mite species typically live 20 days vs 60 days, it would make a difference.

Our response: We appreciate your positive comments. We added female lifespan of T. ludeni in the Material and Method.

Reviewer 4: The authors discuss the implications of the result in terms of pest behavior and outrbreaks. Can there be some implications for management? For example, would early detection be important given that expansion increases with age?

Our response: Thanks. We have added such implication in Discussion.

Reviewer 4: One final point. The authors talk about range expansion. Their results show that the mites increase their dispersal distance as they age and as their density increases. For the highest density tested, the average distance dispersal was close to 4. In this study, that was close to 18 cm. In the field, it is difficult to measure this distance, but how does 18 cm relate to field conditions? Would this mostly allow within or between plant movement? It is understandable that the number of leaf squares is used as a relative distance, but I feel trying to relate that to field conditions would be appropriate.

Our response: Thanks. We have discussed the potential within and between plant movement in relation to dispersal distance in the third and the fifth paragraphs of Discussion.

Reviewer 4: Fig. 3 - can you provide the r-squares for the curves

Our response: We used the curves to demonstrate that with the decrease of egg density over distance, offspring survival increased. We feel that it is not necessary to provide the r-squares.

Reviewer 4:

249: Fig. 4a

251: Fig. 4b-c

257: fig. 3

Our response: Thanks for pointing them out, we revised accordingly except that the L249 was intended to cite Figure 2c.

Reviewer 4: Comments on the Quality of English Language

English is good, I only detected minor grammar issues.

Our response: Thanks. We have proofread it very carefully and corrected the minor issues.

Round 2

Reviewer 3 Report

Comments and Suggestions for Authors

2nd Review of the manuscript “Age and Density of mated females affect dispersal strategies in a spider mite”

            The present manuscript is the revised version where authors have made considerable changes which has improved overall readability of the manuscript. The authors have carefully addressed each every point raised previously and have provided detailed answer to each concern. I agree with the changes authors have made. However, the author’s response to some of my queries reflect a matter of discussion or difference in opinion or completely misunderstanding. I may suggest to move to another round of review, but the during the very short time provided by the Journal (03 days only) it is difficult to debate on each and every point. I leave the decision (Revise or Accept) upon the Editor while I quickly raise very few of the concerns:

Ø  In response to my comment#1

·       The authors have stated “Results generated would facilitate our understanding of the dispersal strategies of T. ludeni and whether factors affecting dispersal probability would also affect dispersal distance.

This is not true in my opinion. The dispersal strategies are not studied so the results would not facilitate that. That’s why I raised question upon design of experimental arenas.

·       The authors replied “We introduced research problems or gaps a few times in introduction,

That’s exactly was my concern. Showing gaps does not mean to state something is unknown. It could be asked like “If we study the age and density dependant dispersal of T. ludeni, what problem we will solve about this species ?” The answer to this question is the need of research statement. Which, still I could not find.

Ø  In response to my comment#2

·       The authors replied “Yes, we mentioned that in general female spider mites disperse under over-crowding and resource depletion conditions, but how different population densities would affect dispersal probability and distance is largely unknown in spider mites.

In my opinion, its not largely unknown. An ecological phenomenon is general and could not go in detail as each spider mite species has a very complex niche and has a very complicated behavior under the effect of biotic and abiotic factors. Hence, when it is known the female spider mites disperse under over-crowding and resource depletion, then the effect of certain population density (as in counts) may not be relevant. Because it depends on variety of other factors affecting coherently.

·       The authors replied “The references 42, 43, 44 investigated the effect of population density and size on reproductive performance of T. ludeni, but it is still not clear the dispersal strategies of the species in response to different population densities.”

If the study of dispersal strategies were of main concern or difference from those references, then my other query on the overall experimental design becomes more relevant and strong. The authors only provided uni-directional dispersal path to mites which in my opinion could not help in studying dispersal strategies.

Ø  In response to my comment “L131-134. 24 hours would be enough to measure the dispersal and laid eggs?”

·       The authors replied “Because we tested the effect of age on dispersal, longer dispersal period would be not appropriate to define female ages.”

I have different opinion here. Its because the treatments made for the age specific dispersal are based on different ages. So it will be pre-defined at the start of the experiment that which treatment has which age female. So my concern was, 24 hours are a short period to study dispersal and here aging would not matter.

Author Response

12 May 2024

Manuscript ID: insects-2980040

Title: Age and density of mated females affect dispersal strategies in a spider mite

Dear Dr. Liu,

Thank you for your decision with one reviewer’s report. We have considered the reviewer’s report very carefully and responded to all his/her comments point by point.

We have highlighted (red) our revisions in the manuscript for your convenience.

Comments from Reviewer 3

Reviewer 3: Comments and Suggestions for Authors

2nd Review of the manuscript “Age and Density of mated females affect dispersal strategies in a spider mite”

The present manuscript is the revised version where authors have made considerable changes which has improved overall readability of the manuscript. The authors have carefully addressed each every point raised previously and have provided detailed answer to each concern. I agree with the changes authors have made. However, the author’s response to some of my queries reflect a matter of discussion or difference in opinion or completely misunderstanding. I may suggest to move to another round of review, but the during the very short time provided by the Journal (03 days only) it is difficult to debate on each and every point. I leave the decision (Revise or Accept) upon the Editor while I quickly raise very few of the concerns:

Our response: We appreciate your positive comments on our revisions. We have responded to your concerns point by point as follows.

Reviewer 3: In response to my comment#1 The authors have stated “Results generated would facilitate our understanding of the dispersal strategies of T. ludeni and whether factors affecting dispersal probability would also affect dispersal distance.” This is not true in my opinion. The dispersal strategies are not studied so the results would not facilitate that. That’s why I raised question upon design of experimental arenas.

Our response: As the reviewer said, we have different opinions on this matter. The dispersal strategies refer to how T. ludeni react to age and density in terms of dispersal. We believe “dispersal strategies” is appropriate to describe the reaction/effect. The term is widely used in literature, for example, the following paper reviewed how different factors would affect dispersal strategies of animals across taxa. As for the experimental arenas, please see our following responses.

Bowler, D.E.; Benton, T.G. Causes and consequences of animal dispersal strategies: relating individual behaviour to spatial dynamics. Biol. Rev. Camb. Philos. Soc. 2005, 80, 205–225.

Reviewer 3: The authors replied “We introduced research problems or gaps a few times in introduction,” That’s exactly was my concern. Showing gaps does not mean to state something is unknown. It could be asked like “If we study the age and density dependant dispersal of T. ludeni, what problem we will solve about this species ?” The answer to this question is the need of research statement. Which, still I could not find.

Our response: Thanks for your patient explanation. T. ludeni is an invasive pest. Investigation into the age and density dependent dispersal of T. ludeni helps our understanding of the invasion mechanisms and provides knowledge for population management and invasion evaluation, which we have discussed in Discussion. We have added this information at the end of the fourth paragraph of Introduction in this revised version.

Reviewer 3: In response to my comment#2

The authors replied “Yes, we mentioned that in general female spider mites disperse under over-crowding and resource depletion conditions, but how different population densities would affect dispersal probability and distance is largely unknown in spider mites.” In my opinion, its not largely unknown. An ecological phenomenon is general and could not go in detail as each spider mite species has a very complex niche and has a very complicated behavior under the effect of biotic and abiotic factors. Hence, when it is known the female spider mites disperse under over-crowding and resource depletion, then the effect of certain population density (as in counts) may not be relevant. Because it depends on variety of other factors affecting coherently.

Our response: Thanks. We agree that each species may have specific dispersal strategies. In the revised manuscript, we briefly mentioned findings of previous studies that spider mites disperse under over-crowding and resource depletion conditions. We then introduced “whether high density would promote the dispersal probability and distance of T. ludeni is unknown”. Please see lines 77-78.

Reviewer 3: The authors replied “The references 42, 43, 44 investigated the effect of population density and size on reproductive performance of T. ludeni, but it is still not clear the dispersal strategies of the species in response to different population densities.

If the study of dispersal strategies were of main concern or difference from those references, then my other query on the overall experimental design becomes more relevant and strong. The authors only provided uni-directional dispersal path to mites which in my opinion could not help in studying dispersal strategies.

Our response: Thanks. We believe that both uni-directional and multiple-directional dispersals have advantages and disadvantages, which we have added a paragraph to discuss these (please see the sixth paragraph of Discussion in the previous revised version). The uni-directional dispersal method is commonly used to test the effect of different factors on dispersal. Please find the following two references using uni-directional experimental design to test dispersal in mites.

Bitume, E.V.; Bonte, D.; Ronce, O.; Bach, F.; Flaven, E.; Olivieri, I.; Nieberding, C.M. Density and genetic relatedness increase dispersal distance in a subsocial organism. Ecol. Lett. 2013, 16, 430–437.

Bowler, D.E.; Benton, T.G. Variation in dispersal mortality and dispersal propensity among individuals: the effects of age, sex and resource availability. J. Anim. Ecol. 2009, 78, 1234-1241.

Reviewer 3: In response to my comment “L131-134. 24 hours would be enough to measure the dispersal and laid eggs?” The authors replied “Because we tested the effect of age on dispersal, longer dispersal period would be not appropriate to define female ages.” I have different opinion here. Its because the treatments made for the age specific dispersal are based on different ages. So it will be pre-defined at the start of the experiment that which treatment has which age female. So my concern was, 24 hours are a short period to study dispersal and here aging would not matter.

Our response: Thanks for your explanation. We set up 24-hour dispersal period for the following reasons. First, it would be more direct and robust to test the effect of age (we started experiment on mites of different ages). Second, we related age-specific dispersal with age-specific reproduction. Allowing dispersal for more time would make recording of age-specific reproduction difficult.

We hope that the revised version is now acceptable for publication in Insects.

Thank you for considering our paper.

Sincerely yours,

Qiao Wang

Professor of Entomology

Massey University

Private Bag 11222 Palmerston North

New Zealand